

# The contribution of energy systems during 15-second sprint exercise in athletes of different sports specializations

Damian Archacki, Jacek Zieliński, Barbara Pospieszna, Michał Włodarczyk and Krzysztof Kusy

Department of Athletics, Strength and Conditioning, Poznan University of Physical Education, Poznan, Poland

## ABSTRACT

**Background**. Long-term adaptations and ongoing training seem to modify the energy system contribution in highly trained individuals. We aimed to compare the energy metabolism profile during sprint exercise in athletes of different specialties.
**Methods**. Endurance ($n = 17$, $20.3 \pm 6.0$ yrs), speed-power ($n = 14$, $20.3 \pm 2.5$ yrs), and mixed ($n = 19$, $23.4 \pm 4.8$ yrs) athletes performed adapted 15-second all-out test before and after a general preparation training period. The contribution of phosphagen, glycolytic, and aerobic systems was calculated using the three-component PCr-LA-O$_2$ method.
**Results**. Between-group differences were observed in the contribution of energy systems in the first and second examinations. The proportions were 47:41:12 in endurance, 35:57:8 in team sports, and 45:48:7 in speed-power athletes. Endurance athletes differed in the phosphagen ($p < 0.001$) and glycolytic systems ($p = 0.006$) from team sports and in the aerobic system from speed-power athletes ($p = 0.003$). No substantial shifts were observed after the general preparatory phase, except a decrease in aerobic energy contribution in team sports athletes ($p = 0.048$).
**Conclusion**. Sports specialization and metabolic profile influence energy system contribution during sprint exercise. Highly trained athletes show a stable energy profile during the general preparation phase, indicative of long-term adaptation, rather than immediate training effects.

## BACKGROUND

Adenosine triphosphate (ATP) resynthesis occurs through three integrated energy systems (*Baker, McCormick & Roberg, 2010*). The phosphagen system relies on phosphocreatine degradation to creatine and the rephosphorylation of adenosine diphosphate to ATP. Along with ATP stored in cells, this system dominates and provides immediate energy in the initial stages (a few seconds) of sprint exercise. There are three main reactions within the phosphagen system: creatine kinase, adenylate kinase, and adenosine monophosphate deaminase reactions. The glycolytic system is based on the restoration of ATP derived from blood glucose and muscle glycogen. Its activity increases in exercise lasting longer

Corresponding author
Damian Archacki,
archacki@awf.poznan.pl

than 10 s through a non-mitochondrial pathway (*Spriet, 1992*). Glycolysis comprises a series of reactions that can be divided into two phases: (i) a preparation or "investment" phase, which requires consuming two ATP molecules, and (ii) a regenerating or "payoff" phase where the net creation of ATP molecules occurs (*Kon, Nakagaki & Ebi, 2019*). In the oxidative or aerobic system, ATP resynthesis occurs through mitochondrial respiration in the presence of oxygen. In this system, free fatty acids and glycogen are the major sources of fuel for restoring energy *via* carbohydrate, lipid, and amino acid oxidation (*Baker, McCormick & Robergs, 2010*; *Gastin, 2001*). During exercise, all pathways contribute to ATP resynthesis through both anaerobic and aerobic processes. However, the extent of their contribution varies based on the type, intensity, and duration of the physical activity. Since various sprint and endurance exercise is characterized by specific metabolism, each sports activity has a specific proportion of the energy systems (*Duffield, Dawson & Goodman, 2004*; *Duffield, Dawson & Goodman, 2005a*; *Duffield, Dawson & Goodman, 2005b*; *Serresse et al., 1988*). In recent years, researchers have estimated the contribution of energy systems in various exercises typical of particular sports disciplines or artistic performance: 400-meter flat and hurdles track races (*Zouhal et al., 2010*), judo (*Julio et al., 2017*), karate (*Beneke et al., 2004*; *Doria et al., 2009*), taekwondo (*Campos et al., 2012*), boxing (*Davis et al., 2015*), jiu-jitsu (*Rodrigues-Krause et al., 2020*), martial arts (*Tortu & Gökhan, 2024*), combat sports (*Tortu, Hazir & Kin Isler, 2024*), team sports (*Tortu et al., 2024*), cross-country skiing (*Losnegard, 2019*), rowing (*De Campos Mello et al., 2009*), kayaking (*Zouhal et al., 2012*), rock climbing (*Bertuzzi et al., 2007*; *Doria et al., 2020*), ballet (*Guidetti et al., 2008*), gymnastics (*Kaufmann et al., 2022b*), badminton (*Fu et al., 2021*), swimming (*Pelarigo et al., 2017*; *Shimoyama, Tomikawa & Nomura, 2003*), soccer (*Ulupınar et al., 2021*; *Ulupınar, Hazır & Kin İşler, 2023*), table tennis (*Milioni et al., 2018*), and exergames (*Park et al., 2020*; *Yang et al., 2022*).

Despite substantial research, there is a notable lack of studies examining sprinting efforts. Sprinting is crucial in competitive sport, not only in speed-power, but also in endurance disciplines due to, for example, acceleration, overtaking, and finishing (*Gastin, 2001*). On the other hand, aerobic metabolism contributes to some extent to the total energy supply even during all-out sprint exercise. It was found that in the 100- and 200-meter runs, the contribution of the aerobic system in sprinters was 8.9 ± 3.3% and 20.7 ± 8.5%, respectively (*Duffield, Dawson & Goodman, 2004*). In rugby players, about 20% of the energy turnover in the standard 30-second Wingate test was derived from aerobic metabolism (*Beneke, Pollman & Hutler, 2002*). Medbo and Tabata showed that aerobic processes in the 30-second exhaustive cycling provided about 40% of the total energy release (*Medbo & Tabata, 1989*). Huegen et al. reported that during 12–22-second sprinting aerobic system contribution was about 30% (*Haugen, McGhie & Ettema, 2019*; *Haugen et al., 2019*). In world-class sprinters, the average time to cover 100 m ranges from 9.58 s to 10.20 s, and over 200 m from 19.19 s to ~20 s (*Haugen, McGhie & Ettema, 2019*). Importantly, the average sprint in field-based team sports is much shorter than 20 s (*Spencer et al., 2005*). Several authors indicated that single high-intensity sprint runs last only 2–4 s in soccer (*Andrzejewski et al., 2018*; *Spencer et al., 2005*) and ~4 s in elite field hockey players (*Spencer et al., 2004*). In line with this, the average distance of a single

sprint in elite soccer is 10–20 m, depending on the player's position (*Andrzejewski et al., 2013*; *Andrzejewski et al., 2015*). The analysis of UEFA Europa League matches revealed that ~90% of all sprinting activities lasted less than 5 s and, consequently, sprints longer than 5 s accounted only for ~10% (*Andrzejewski et al., 2013*). In this context, calculating the energy systems contribution in sprint exercises lasting less than the typical 'anaerobic' 30-second Wingate test may be more relevant and useful from the practical point of view.

The results of previous reports (*Bertuzzi et al., 2007*; *Zouhal et al., 2012*) suggest that endurance athletes, when performing sports-specific exercise, utilize the aerobic system to a greater extent due to their better-developed aerobic capacity, while speed-endurance athletes rely more heavily on anaerobic energy sources, in line with their adaptations to sprint exercise (*Spencer & Gastin, 2001*). Thus, it seems that in athletes of various metabolic profiles, the type of metabolism that is better developed in the process of long-term training adaptation is more active. Investigating whether there are sport-related differences in energy systems contribution and whether such energy profiles are modifiable can help gain a better understanding of the specific training effects on exercise metabolism. In practical terms, this can support better personalization of training strategies in high-performance athletes and other physically active people. To the best of our knowledge, no study has yet been published that directly compares energy system contributions in individuals representing divergent training profiles performing the same standardized test. It is also unknown whether the energy systems contribution in professional athletes is influenced by the training process. The question arises, thus, whether the "energy profile" of the competitive athlete is rather unmodifiable and typical of the sport, or whether it changes in response to ongoing training.

Accordingly, this study aimed to estimate and compare the contribution of three energy systems during a 15-second all-out sprint exercise test in athletes with different sport-related physiological adaptations. Our first hypothesis is that the contribution of energy systems during sprint exercise is determined by specific long-term exercise adaptations related to sport specialization. The second hypothesis is that the contribution of energy systems in athletes changes in response to the ongoing training process. Our findings may equip coaches and sports scientists with specific insights for categorizing athletes, aiding in the appropriate discipline selection during various stages of sports training. The evaluation of energy system contributions during the 15-second all-out test could serve as an additional diagnostic tool for sport-specific selection to identify predispositions.

## METHODS

### Participants

Portions of this text were previously published as part of a preprint (*Archacki et al., 2023*).

Initially, the study encompassed 102 athletes from various disciplines who were tested in the general preparation period (Fig. 1). Due to the insufficient number of females in the team sports group and/or non-attendance at the second examination, 50 participants were included in the analysis. Once the data was collected, athletes were assigned to one of three groups: endurance, speed-power, and team sports according to their training specificity

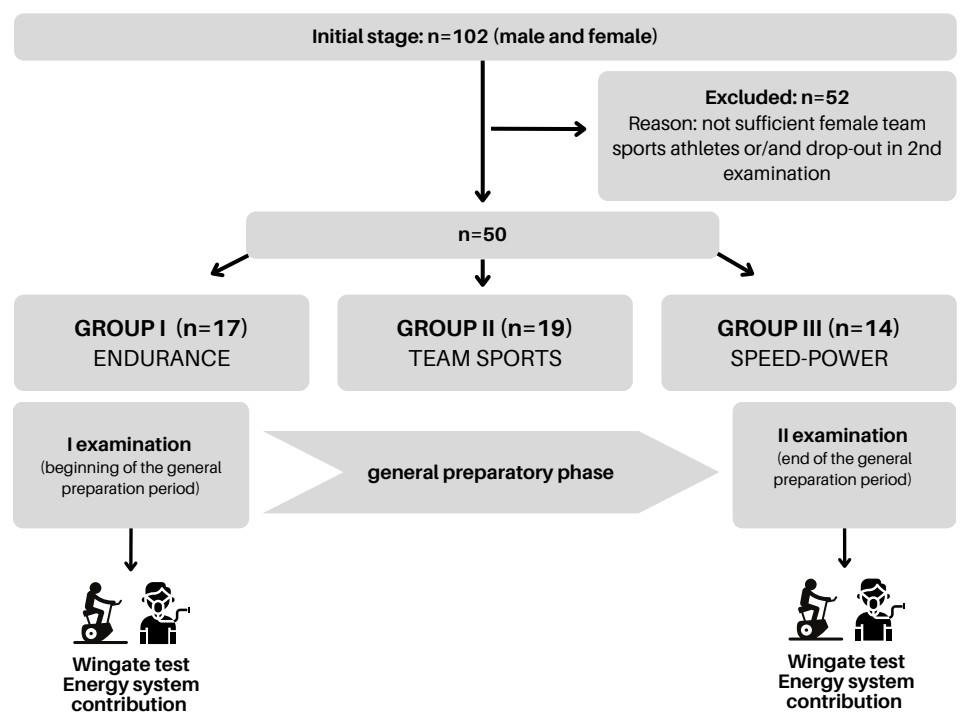

**Figure 1** **Schematic representation of the study design.** Graph created using Canva.

which resulted from long-term adaptations. The groups were composed to present equal levels of peak and mean power per kilogram of body mass and skeletal muscle mass. As a result of this procedure, we included three athletic groups: the endurance group ($n = 17$) consisted of triathletes (specializing in 1.5 km swimming, 40 km cycling, and 10 km running) and long-distance runners (specializing in 5 and 10 km), both competing at the national level aged $20.3 \pm 6.0$ years, body mass $68.2 \pm 6.2$ kg, body fat $14.2 \pm 3.3\%$, height $176.7 \pm 5.0$ cm. The speed-power group ($n = 14$) consisted of sprinters (competing in the 60–200 m distances) and Olympic taekwondo athletes, both competing at the national level aged $20.3 \pm 2.5$ years, body mass $78.2 \pm 9.2$ kg, body fat $15.7 \pm 4.8\%$, height $181.8 \pm 5.7$ cm. The team sports group ($n = 19$) consisted of amateur soccer players, aged $23.4 \pm 4.8$ years, body mass $75.2 \pm 7.1$ kg body fat $17.7 \pm 4.9\%$, height $180.1 \pm 5.5$ cm. The investigated three male groups represented distinctly different metabolic and physiological adaptations to observe whether specific long-term training adaptations would result in different energy system responses. Sprinters and taekwondo athletes, using mainly high-intensity power-based activities during sports competitions, represented the speed-power group. Long-distance runners and triathletes, whose competitive activity mostly relies on low- to moderate-intensity exercise, represented the endurance group. Soccer players, from whom both speed and endurance efforts are required during the match, formed the team sports group.

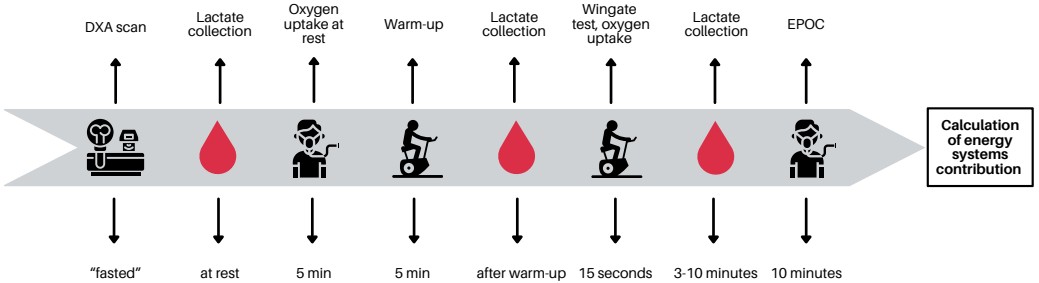

**Figure 2** **Testing procedure.** Graph created using Canva.

## Study design

Prior to the primary examination, athletes underwent a familiarization session with the procedures and testing protocols. The first examination was performed after a transition (detraining) phase and the second at the end of the general preparatory phase (Fig. 1). The procedures are described in detail in subsequent sections. The endurance group engaged in training sessions 6–7 days per week, including 1–2 strength and conditioning sessions, with an average session duration of 120 min, and a total volume of ~182 training hours. Similarly, the athletes in the speed-power group undertook 6–7 training sessions per week, including 2 sessions of resistance training oriented towards building general strength, with an average duration of 90 min per session and a total volume of ~137 training hours. The team sports group conducted 3–4 training sessions per week, with an average session duration of 80 min and a total volume of ~65 training hours, but they did not complete any resistance training outside of soccer training. In addition to the criterion of sports specialization, participants were categorized into groups based on medians for maximal oxygen uptake ($\dot{V}O_{2max}$) and Peak Power (PP). Consequently, each athlete was assigned to a low- or high-$\dot{V}O_{2max}$ group, a low- or high-PP group, and a combination thereof.

## Procedures

All tests were performed in the Human Movement Laboratory of the Department of Athletics and Strength and Conditioning at the Poznan University of Physical Education. The project was approved by the Ethics Committee at the Poznan University of Medical Sciences on September 9, 2020 (decision No 627/20) and conducted following the principles of the Helsinki Declaration. All participants were familiarized with the aim, procedures, and risks of the study and signed written informed consent.

Athletes were asked to refrain from high-intensity or long-duration training sessions 24–48 h before the testing. The participants were instructed to be fasted before body composition analysis and thereafter they ate a light meal. The blood samples were collected before and after the 15-s all-out test. The room temperature was maintained at 20–21 °C. Testing procedures are presented in Fig. 2.

## Body composition

A digital stadiometer (SECA 285; SECA, Hamburg, Germany) was used to measure height and weight. Body composition analysis was assessed by dual X-ray absorptiometry (DXA)

method with Lunar Prodigy Pro device (GE Healthcare, Madison, WI, USA) and enCORE v16 SP1 software. Prior to each measurement session, the device was calibrated with a phantom in accordance with the manufacturer's guidelines. During the examination, subjects were instructed to wear only their underwear and to remove all jewelry and metal objects to minimize measurement error. A single trained technician performed all DXA scans that were analyzed according to the protocol proposed by *Nana et al. (2015)*. The variables of interest were total body fat, leg lean mass, and total-body skeletal muscle mass (SMM), the latter calculated according to the method proposed by *Kim et al. (2002)*.

### 15-s all-out test

All participants were familiarized with the test protocol before the study. The test was conducted on the cycle ergometer Cyclus 2 (RBM elektronik-automation GmbH, Leipzig, Germany) according to the recommendations for standardization (*Inbar, Bar-Or & Skinner, 1996*). The protocol started with a 5-minute warm-up at each athlete's own pace, during which a power output of 25 W was maintained for one minute and then increased to 50 W for the last 4 min, interspersed with a few short sprints lasting up to 5 s. Then, after a ~2 min rest, the subjects were instructed to pedal as fast as possible for 15 s. The precondition for starting the test was blood lactate concentration below 3 mmol $L^{-1}$, and in the case of an elevated level, the start of the test was postponed until the desired value was reached. The load was set at 0.085 kg per kg body weight (*Katch et al., 1977*). Strong verbal encouragement was given each time throughout the test. The subjects were required to remain seated after the test for 10 min. The peak power was defined as the highest mechanical power achieved in a 5-second segment of the test, whereas the mean power was determined by averaging the instantaneous power values throughout the 15-second test period. The fatigue index indicated the rate at which power declined from the highest value (peak power; PP) to the 5-second period with the lowest values after reaching PP. Fatigue index was calculated as the difference between the PP and the minimum power divided by the time of the power drop (*Bar-Or, 1987*; *Coppin et al., 2012*).

### Exercise testing

The respiratory parameters were measured by MetaMax 3B-R2 and analyzed using MetaSoft Studio Software (both: Cortex Biophysik GmbH, Leipzig, Germany). Respiratory data were collected 5 min before, throughout, and 10 min after the all-out test. To determine lactate concentration, blood samples were collected from the fingertip at rest (5 min before starting warm-up), after warm-up, and every minute during the 3–10 min period after the test. The resting and peak post-exercise values were used for analysis. Biosen C-line (EKF diagnostic GmbH, Barleben, Germany) was used to measure lactate levels using capillary whole blood. The pilot study indicated that the peak lactate concentration occurred three minutes after the test at the earliest, so we refrained from collecting blood immediately after the test.

Furthermore, following the 15-s all-out test, all athletes participated in an incremental exercise test to exhaustion on the h/p Cosmos Pulsar treadmill (Sports & Medical GmbH, Nussdorf-Traunstein, Germany) to assess $\dot{V}O_{2max}$. A continuous ramp protocol was utilized. In brief, athletes stood still for 3 min, then started to walk at the speed of 4 km $h^{-1}$
for 3 min (adaptation), after which the speed was gradually and smoothly increased by an average of 0.67 km h$^{-1}$ min$^{-1}$ until individual voluntary exhaustion was reached. It was considered that $\dot{V}O_{2max}$ was reached if at least three of the following criteria were met: (i) plateau for $\dot{V}O_{2max}$, (ii) respiratory exchange ratio $\geq 1.10$, (iii) heart rate close to an individual's maximum ($\pm 5\%$), (iv) blood lactate $\geq 9$ mmol/L, and (v) perceived exertion severity $\geq 17$ on the Borg scale.

## Calculation of the energy system contributions

We based the calculation on the PCr-LA-O$_2$ method and procedures described by *Beneke et al. (2004)* and *Bertuzzi et al. (2007)* to estimate aerobic, glycolytic, and phosphagen energy system contributions (*Beneke, Pollman & Hutler, 2002*). The method is considered reliable due to its ability to capture the dynamic metabolic changes that occur during such intense exercise (*Kaufmann et al., 2022a*). Additionally, *Li et al. (2015)* have suggested that the Maximal Accumulated Oxygen Deficit (MAOD) method tends to overestimate the aerobic contribution compared to PCr-LA-O$_2$ (*Li et al., 2015*).

We expressed the energy expenditure in kilojoules for the three systems, assuming the caloric equivalent of 20.9 kJ per 1 liter O$_2$. Energy derived from the aerobic system (E$_{AER}$) was calculated by subtracting $\dot{V}O_2$ at rest (5 min) from the $\dot{V}O_2$ during maximal exercise using the trapezoidal method (*Beneke, Pollman & Hutler, 2002*). Energy from the glycolytic system (E$_{LA}$) was calculated by using the oxygen equivalent from blood lactate concentration; it was estimated as the difference between peak lactate concentration and lactate at rest. The net value of 1 mmol L$^{-1}$ was considered as the equivalent to 3 mLO$_2$ kg body mass$^{-1}$ (*Spencer et al., 2004*). The energy from the phosphagen system (E$_{PCR}$) was calculated from the fast component of the excess post-exercise oxygen consumption (EPOC) (*Gaesser & Brooks, 1984*). The $\dot{V}O_2$ consumption data were collected during 10 min of recovery immediately after the test according to procedures suggested by *Bertuzzi et al. (2016)* and calculated using the bi-exponential model (*Özyener et al., 2001*). The following formulas were used:

$$\dot{V}O_{2(t)} = \dot{V}O_{2baseline} + A_f[e^{-(t-td)/\tau_f}] + A_s[e^{-(t-td)/\tau_s}]$$

$$E_{PCR} = A_f \, \tau_f,$$

where $\dot{V}O_{2(t)}$ is the oxygen uptake at time $t$, $\dot{V}O_{2baseline}$ is the oxygen uptake at rest, A is the amplitude, td is the time delay, $\tau$ is the time constant, $f$ denotes the fast, and $s$ denotes the slow component of EPOC.

To achieve a sufficiently high fit of each EPOC kinetic curve, we tested calculations at various times of oxygen consumption from the 3rd to 10th minute of recovery. Due to the most reliable and consistent results, the 7-minute EPOC period was taken into the analysis. This process resulted in obtaining high coefficients of determination for all kinetic curves analyzed ($r^2 = 0.80$–$0.98$). The time we used to analyze EPOC is consistent with other studies that also utilized recovery time from 6th to 10th minutes after various high-intensity exercises *e.g.*: 100-meter sprint, 15-second, and 30-second all-out cycling (*Artioli et al., 2012*; *Doria et al., 2020*; *Park et al., 2021*; *Yang et al., 2023*). Also, *Bertuzzi et al. (2016)* indicated that measuring the first 6 min of the $\dot{V}O_2$ recovery after both moderate- and high-intensity exercise is sufficient to determine the phosphagen system. Additionally,

raw data was edited as proposed by *Lamarra et al. (1987)* and *Myers et al. (1990)*. Hence, values greater than 3 standard deviations were omitted to exclude deviations caused by coughing, sighing, *etc*. The data was analyzed using the GEDAE-LaB software, developed by researchers from Sao Paulo University and the Federal University of Pernambuco, Brazil (*Bertuzzi et al., 2016*). The total energy expenditure ($E_{TOT}$) was calculated as the sum:

$$E_{TOT} = E_{AER} + E_{LA} + E_{PCR}$$

The contribution of each energy system was calculated as the percentage of $E_{TOT}$.

## Statistical Analyses

Data distribution normality was verified using the Shapiro–Wilk test. Since the distribution was normal, all values were presented as mean and standard deviation. A two-way repeated measures analysis of variance (ANOVA) was performed to test the effects of the athletic group, examination, and their interaction for each energy system. One-way ANOVA was used to detect differences between energy systems in each separate group of participants. Levene's test was used to assess the homogeneity of variance. The Bonferroni *post-hoc* test was conducted if significant main effects were found. The partial eta squared ($\eta^2$) was calculated as a measure of the effect size and interpreted as small (0.01), medium (0.06), or large (0.14) (*Cohen, 1988*). All statistical analyses were performed using the Statistica software package (TIBCO Software Inc., Palo Alto, CA, USA). The significance level was set at $p < 0.05$.

## RESULTS

### Body composition and aerobic capacity

The speed-power group had the highest and the endurance group had the lowest SMM and leg lean mass. The team sports group represented the middle value of both SMM and leg lean mass and did not differ significantly from the speed-power and endurance groups. Team sports athletes significantly increased their SMM and leg lean mass and decreased the percentage of body fat between examinations. The endurance and speed-power groups did not significantly change these components during the study period. Endurance athletes had significantly higher absolute $\dot{V}O_{2max}$ than team sports athletes and higher relative $\dot{V}O_{2max}$ than both speed-power and team sports groups. Additionally, the team sports group showed a significant increase in both absolute and relative $\dot{V}O_2$max between examinations, whereas other groups did not (Table 1).

### Peak and mean power

Speed-power athletes had higher absolute PP and mean power and fatigue index than the endurance group. The between-group differences in relative power (both per kg of total body mass and SMM) were insignificant. There were also no significant changes in any of the above variables between the two examinations (Table 2).

### Within-group differences in energy system contribution

Both, the percentage contribution and absolute energy expenditure (kJ), were significantly different between the three energy systems in most defined groups. The contribution of the

**Table 1  Changes in body components and aerobic capacity between the first (PRE) and second (POST) examination.** Values are expressed as means and standard deviations ($p < 0.05$).

| | ENDURANCE | | TEAM SPORTS | | SPEED-POWER | | ANOVA effects | | | | | |
| | | | | | | | Group | | Examination | | Group* examination | |
| | PRE | POST | PRE | POST | PRE | POST | $p$ | $\eta^2$ | $p$ | $\eta^2$ | $p$ | $\eta^2$ |
|---|---|---|---|---|---|---|---|---|---|---|---|---|
| BM (kg) | $68.3 \pm 6.3^*$ | $68.7 \pm 5^*$ | $75.2 \pm 7.1$ | $75.4 \pm 6.8$ | $78.1 \pm 9.2$ | $78.8 \pm 9.7$ | 0.001 | 0.241 | 0.087 | 0.060 | 0.711 | 0.014 |
| BF (%) | $14.2 \pm 3.3$ | $14.2 \pm 3.1$ | $17.7 \pm 4.9$ | $16.5 \pm 4^{\#}$ | $15.7 \pm 4.8$ | $15.3 \pm 4.9$ | 0.118 | 0.086 | 0.031 | 0.094 | 0.060 | 0.112 |
| SMM (kg) | $29.7 \pm 2.9^*$ | $29.7 \pm 2.6^*$ | $32.1 \pm 3.3$ | $32.7 \pm 3.4^{\#}$ | $34.7 \pm 4.8$ | $35.3 \pm 4.6$ | <0.001 | 0.263 | <0.001 | 0.197 | 0.025 | 0.144 |
| LLM (kg) | $19.2 \pm 1.8^*$ | $19.1 \pm 1.7^*$ | $21 \pm 2.1$ | $21.4 \pm 2.3^{\#}$ | $22.7 \pm 3.0$ | $22.9 \pm 2.9$ | 0.002 | 0.294 | 0.014 | 0.120 | 0.007 | 0.187 |
| $\dot{V}O_2$max (L min$^{-1}$) | $4.2 \pm 0.4^{\dagger}$ | $4.4 \pm 0.4^{\dagger}$ | $3.6 \pm 0.2$ | $3.8 \pm 0.3^{\#}$ | $4.1 \pm 0.6$ | $4.3 \pm 0.6$ | <0.001 | 0.330 | 0.003 | 0.188 | 0.923 | 0.003 |
| $\dot{V}O_2$max (mL min$^{-1}$ kg$^{-1}$) | $62.2 \pm 4.5^{\dagger *}$ | $64.9 \pm 7.3^{\dagger *}$ | $49.3 \pm 3.9$ | $51.1 \pm 4.8^{\#}$ | $51.8 \pm 6.4$ | $53.4 \pm 7.9$ | <0.001 | 0.585 | 0.008 | 0.155 | 0.775 | 0.012 |

Notes.
[#]Significantly different from the first examination.
[*]Significantly different from speed-power athletes at the same examination.
[†]Significantly different from mixed athletes at the same examination.
  BM, body mass; BF, body fat; SMM, skeletal muscle mass; LLM, leg lean mass; $\dot{V}O_2$max, maximal oxygen uptake.

**Table 2  The change in mechanical variables of the Wingate test between the first (PRE) and second (POST) examination.** Values are expressed as means and standard deviations ($p < 0.05$).

| | ENDURANCE | | TEAM SPORTS | | SPEED-POWER | | ANOVA effects | | | | | |
| | | | | | | | Group | | Examination | | Group*examination | |
| | PRE | POST | PRE | POST | PRE | POST | p | $\eta^2$ | p | $\eta^2$ | p | $\eta^2$ |
|---|---|---|---|---|---|---|---|---|---|---|---|---|
| PP (W) | $784 \pm 95^*$ | $772.4 \pm 88^*$ | $853 \pm 123$ | $860.8 \pm 101$ | $918 \pm 106$ | $932 \pm 143$ | <0.001 | 0.244 | 0.713 | 0.002 | 0.536 | 0.026 |
| PP (W kg BM$^{-1}$) | $11.5 \pm 0.9$ | $11.3 \pm 0.8$ | $11.3 \pm 1$ | $11.4 \pm 0.9$ | $11.6 \pm 1.1$ | $11.8 \pm 1.2$ | 0.280 | 0.052 | 0.735 | 0.002 | 0.528 | 0.026 |
| PP (W kg SMM$^{-1}$) | $26.4 \pm 1.9$ | $26 \pm 1.9$ | $26.6 \pm 2.2$ | $26.3 \pm 1.3$ | $26.7 \pm 3.6$ | $26.4 \pm 2.1$ | 0.843 | 0.007 | 0.424 | 0.013 | 0.987 | 0.000 |
| MP (W) | $698 \pm 78.7^*$ | $697 \pm 76^*$ | $727.6 \pm 72$ | $750.5 \pm 81$ | $798.3 \pm 82$ | $806.4 \pm 113$ | 0.003 | 0.216 | 0.148 | 0.043 | 0.324 | 0.046 |
| MP (W kg BM$^{-1}$) | $10.2 \pm 0.7$ | $10.2 \pm 0.7$ | $9.7 \pm 0.7$ | $10.0 \pm 0.7$ | $10.1 \pm 0.9$ | $10.2 \pm 0.9$ | 0.194 | 0.067 | 0.658 | 0.004 | 0.237 | 0.057 |
| MP (W kg SMM$^{-1}$) | $23.5 \pm 1.6$ | $23.5 \pm 1.7$ | $22.7 \pm 1$ | $23 \pm 1.1$ | $23.3 \pm 3.1$ | $22.9 \pm 1.5$ | 0.410 | 0.037 | 0.874 | 0.000 | 0.577 | 0.023 |
| FI (W s$^{-1}$) | $14.9 \pm 3.7$ | $14.9 \pm 5.9^*$ | $21.8 \pm 9.3$ | $21.6 \pm 8.2$ | $21.5 \pm 5.2$ | $24.7 \pm 9.2$ | <0.001 | 0.255 | 0.362 | 0.017 | 0.400 | 0.038 |
| LA$_{rest}$ (mmol/L$^{-1}$) | $2.1 \pm 0.5$ | $1.9 \pm 0.4$ | $1.9 \pm 0.5$ | $1.5 \pm 0.6$ | $2.1 \pm 0.6$ | $1.9 \pm 0.6$ | 0.165 | 0.073 | 0.002 | 0.180 | 0.449 | 0.033 |
| LA$_{peak}$ (mmol/L$^{-1}$) | $8.3 \pm 1.7$ | $8.0 \pm 1.3$ | $9.8 \pm 2.1$ | $9.6 \pm 1.6$ | $9.6 \pm 1.8$ | $9.8 \pm 1.7$ | 0.009 | 0.180 | 0.410 | 0.014 | 0.499 | 0.029 |

Notes.
[*]Significantly different from speed-power athletes at the same examination
  PP, body mass; MP, mean power; FI, fatigue index; SMM, skeletal muscle mass; LA$_{rest}$, lactate concentration at rest; LA$_{peak}$, peak lactate concentration.

phosphagen system did not differ significantly from glycolytic in the speed-power group, in athletes with high $\dot{V}O_{2max}$, and athletes with high PP. In all groups, the contribution of the oxygen system was the lowest (Figs. 3A–3D and 4A–4D).

## Between-group differences in energy system contribution

Between-group differences were observed in the percentage contribution of the phosphagen and glycolytic systems between the endurance and team sports group (Fig. 3A), and athletes with high $\dot{V}O_{2max}$ (Fig. 3C). The aerobic system differed significantly between the endurance and speed-power groups in the second examination (Fig. 3A), and between athletes with high PP and low PP (Fig. 3D). Between-group differences in absolute energy expenditure (kJ) were observed only for the glycolytic system (Figs. 4A–4D).

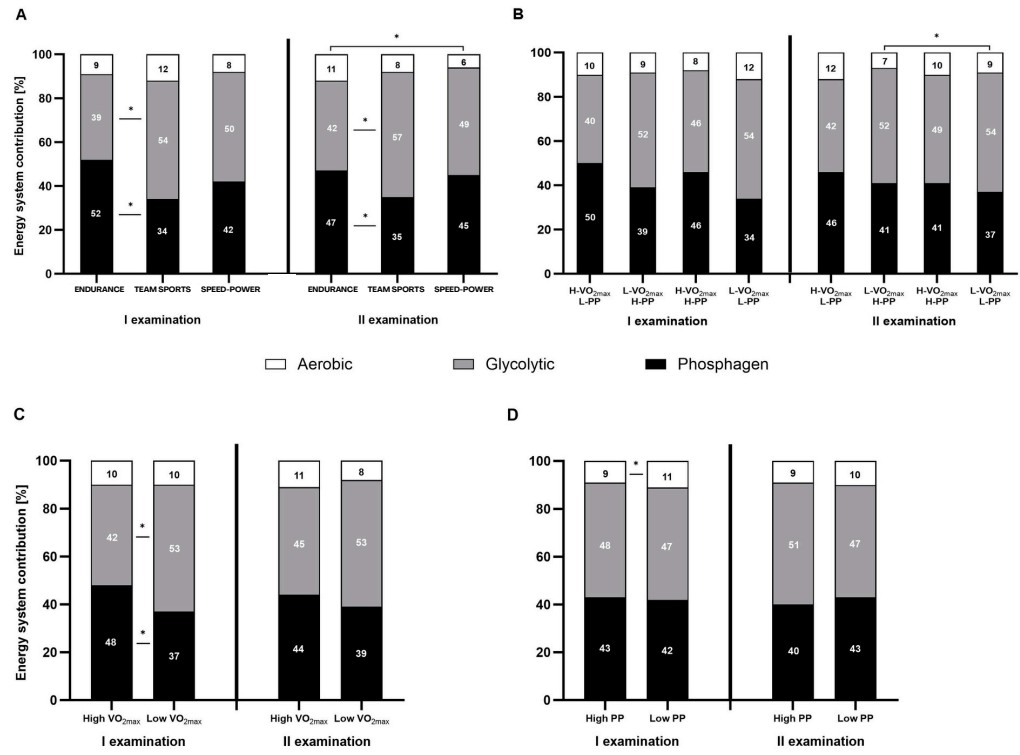

**Figure 3  Relative contribution (%) of the energy systems during the 15-second Wingate test.** (A) Relative contribution (%) in endurance, mixed, and speed-power athletes. (B) Athletes grouped by high/low maximal oxygen uptake $\dot{V}O_{2max}$ and high/low peak power (PP). (C) Athletes grouped by high/low $\dot{V}O_{2max}$. (D) Athletes grouped by high/low high/low PP. Black: phosphagen system, grey: glycolytic system, white: aerobic system. Values are expressed as means of each energy system (%). ⋆ significant differences in the contribution of various systems between groups. H-$\dot{V}O_{2max}$/L-PP–High $\dot{V}O_{2max}$ and low peak power L-$\dot{V}O_{2max}$/H-PP–Low $\dot{V}O_{2max}$ and high peak power H-$\dot{V}O_{2max}$/H-PP–High $\dot{V}O_{2max}$ and high peak power L$\dot{V}O_{2max}$/L-PP–Low $\dot{V}O_{2max}$ and low peak power.

### Between-examination differences in energy system contribution

The only significant difference between the first and second examinations was found in the team sports group, in which the proportion of the aerobic system decreased by 2% (Fig. 4A). Detailed results and information on within-group, between-group, and between-examination differences are provided in Tables S1–S4.

## DISCUSSION

This is the first study to compare the response of three metabolic energy systems during sprint exercise before and after a training period among athletes of different disciplines. The main findings are that (i) significant differences exist between relative energy system contribution and the energy expenditure depending on the sport and metabolic profile, and (ii) there is a lack of clear influence of the ongoing training process on the energy system contribution. Thus, the results indicate that mainly specific long-term training adaptations induced through sports training (discipline) determine energy system contribution in highly trained athletic cohorts.

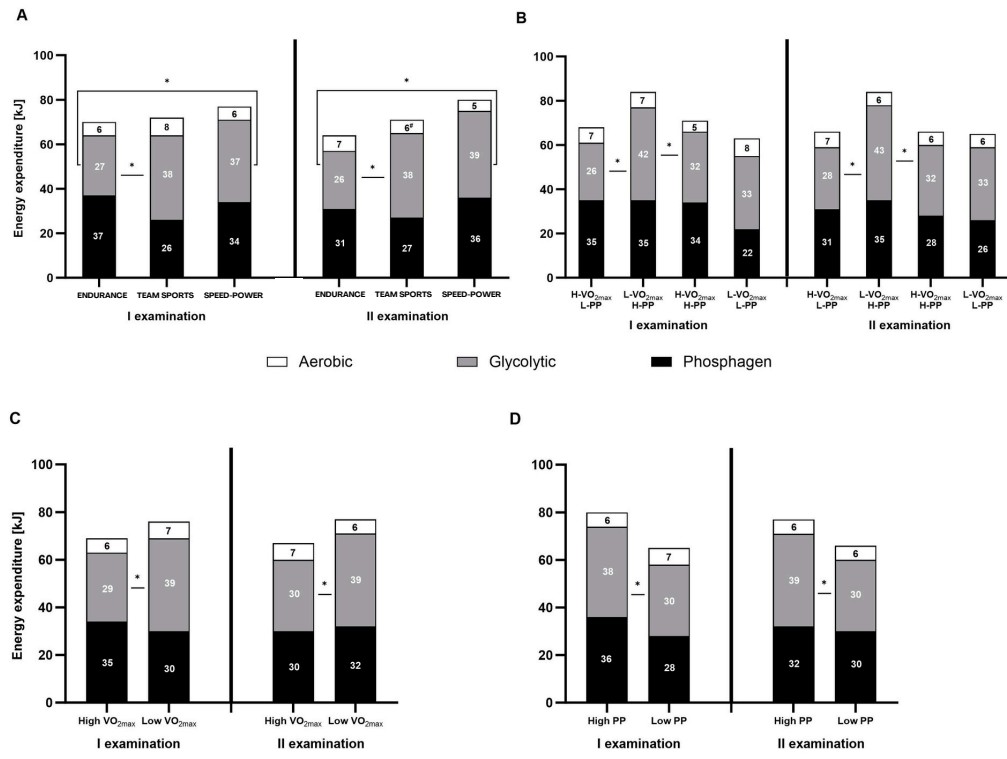

**Figure 4** **Energy expenditure (kJ) during the 15-second Wingate test.** (A) Energy expenditure (kJ) in endurance, mixed, and speed-power athletes. (B) Athletes grouped by high/low maximal oxygen uptake $\dot{V}O_{2max}$ and high/low peak power (PP). (C) Athletes grouped by high/low $\dot{V}O_{2max}$. (D) Athletes grouped by high/low PP. Black: phosphagen system, grey: glycolytic system, white: aerobic system. Values are expressed as means of each energy system (kJ). * significant differences in energy expenditure of various systems between groups. H-$\dot{V}O_{2max}$/L-PP–High $\dot{V}O_{2max}$ and low peak power L-$\dot{V}O_{2max}$/H-PP–Low $\dot{V}O_{2max}$ and high peak power H-$\dot{V}O_{2max}$/H-PP–High $\dot{V}O_{2max}$ and high peak power L-$\dot{V}O_{2max}$/L-PP–Low$\dot{V}O_{2max}$ and low peak power.

In earlier studies, the contribution of energy systems (phosphagen: glycolytic: aerobic) during the standard 30-s Wingate test was 31: 50: 18% and 32: 45: 23% among rugby players (*Beneke, Pollman & Hutler, 2002*) and judo athletes (*Julio et al., 2019*), respectively. Despite the fact, that the duration of the Wingate test was longer compared to our investigation, the results of those studies were similar to our mixed and speed-power groups. The substantially lower contribution of the aerobic system in our athletes was arguably due to the shorter duration of the test. In general, the glycolytic system predominates and the phosphagen system remains second in the speed-power or resistance groups, indicating specific training-related adaptations. The average contribution of the aerobic system ranged from 8 to 12% among our participants, which is consistent with other studies that showed values in the range of 5–18% during sprint exercise (6–20 s of cycling) (*Bogdanis et al., 1998*; *Nummela et al., 1996*). In all our groups, the combined anaerobic energy contribution was about 90% (phosphagen + glycolytic). This corresponds with the results of an earlier study by Gastin (*Gastin, 2001*), who reported that during maximal 15-second exercise about 88–90% of energy was derived from anaerobic metabolism.

## Differences between athletic groups in energy system contribution

Paradoxically, we did not confirm that speed-power athletes use anaerobic metabolism to a greater extent than endurance athletes as was hypothesized. In a certain sense, this finding may disrupt existing beliefs and stereotypes, that athletes with an endurance-type adaptation do not utilize the phosphagen system to such an extent as those with speed-power and mixed adaptations. Surprisingly, we observed that the endurance group showed a significantly higher contribution of the phosphagen system (both absolute and percentage) compared to the mixed and speed-power groups. To the best of our knowledge, there is only one former study on energy system contribution in endurance athletes performing sprint exercise. *Granier et al. (1995)* compared sprinters and middle-distance runners in the 30-s Wingate test. They estimated the anaerobic: aerobic systems ratio to be 55:45% for endurance runners and 72:28% for sprinters, whereas in our speed-power and endurance athletes the combined contribution of anaerobic metabolism was as high as ∼91%. Also, in contrast to our study, they did not obtain significant differences between the athletic groups. This discrepancy may be due to, among other things, longer test duration and vastly different methods to estimate the energy systems contribution, with only two main systems available in the research by *Granier et al. (1995)*. According to *Serresse et al. (1988)*, it is essential to dissociate anaerobic lactic and alactic (phosphagen) metabolism indicators. *Shimoyama, Tomikawa & Nomura (2003)* examined endurance- and sprint-type swimmers, specializing in the distances of 100 and 200 m, during continuous *vs* 10-second interval swimming with different recovery times. It was found that the anaerobic contribution was significantly higher in endurance-type athletes than those of sprint type. These findings are in agreement with the study of *Jansson et al. (1990)*, who indicated that athletes with a higher aerobic capacity might have a better ability for phosphocreatine resynthesis during interval exercise. They concluded that athletes with lower aerobic capacity are more likely to rely upon anaerobic ATP resynthesis *via* glycolysis. Thus, the earlier and our research suggest, that endurance-type athletes tend to utilize the phosphagen system to a greater extent than the glycolytic system in sprint exercise. In addition, endurance training increases capillary density, mitochondrial content, and oxidative enzyme activity, reducing the need for glycolysis during high-intensity exercise (*Green, Fraser & Ranney, 1984*). Moreover, there appears to be a potential association with $\dot{V}O_{2max}$ levels, as our results indicate. The percentage contribution of the phosphagen system was dominant in the high-$\dot{V}O_{2max}$ group and the glycolytic system in the low-$\dot{V}O_{2max}$ group. Also, for absolute energy expenditure (kJ), differences in the glycolytic system were evident between the high- and low-$\dot{V}O_{2max}$ groups.

The contribution of the glycolytic energy was greater in speed-power and mixed (∼50% and ∼54%, respectively) athletes than in the endurance group (∼38%). One explanation may be that, in general, speed-power and team sports athletes have a greater ability to produce higher exercise-induced blood lactate concentration (*Draper & Wood, 2005*; *Granier et al., 1995*; *Torok et al., 1995*). In addition, a higher percentage of fast-twitch fibers, characteristic of sprinters, is associated with higher exercise-induced blood lactate concentrations (*Froese & Houston, 1987*). The high level of lactate shows the ability to produce energy from anaerobic lactic metabolism *via* glycolysis (*Beneke, Pollman & Hutler,*

*2002*; *Di Prampero & Ferretti, 1999*). One can assume that, due to a lower capacity to produce and utilize lactate in the process of glycolysis, endurance athletes rely more on ATP resynthesis *via* the phosphagen system during sprint exercise, during which aerobic metabolism, their greatest asset, is only minimally activated. When glycolytic pathways are less developed and oxygen pathways cannot be fully utilized, the phosphagen system 'replaces' them to some extent as a more efficient and available energy source during very short exercise of maximum intensity. Sprinters typically possess a greater amount of fast-twitch fibers, beneficial for short, high-intensity efforts. In contrast, endurance athletes have a higher percentage of slow-twitch fibers, which are more effective for prolonged, low-intensity activities (*Costill et al., 1976*; *Jansson et al., 1990*; *Saltin & Gollnick, 2011*).

Moreover, the fast-twitch (FT) motor units in white skeletal muscle (FTb) are best suited for high-intensity, short-duration activities, primarily using glycolytic degradation of glycogen to lactate for ATP production. Consequently, during high-intensity exercise, the depletion of glycogen from FTb units is greater compared to slow-twitch (ST) units (*Saltin & Gollnick, 2011*). Thus, athletes with long-term endurance adaptations, characterized by a predominance of ST fibers, are more likely to regenerate ATP *via* the phosphagen pathway rather than glycolysis during a short sprint. The mixed and endurance groups showed discrepancies in the contribution of the phosphagen and glycolytic systems. The glycolytic system dominated in the team sports group, whereas the contribution of the two systems remained similar in speed-power athletes. This indicates that the two components of the anaerobic metabolism are more balanced in the speed-power group. This may be related to specific long-term adaptation including sprint exercise of similar duration as the test we applied, most often interspersed by full recovery intervals. The team sports group consisted of soccer players, whose competitive match plays require the ability to repeatedly perform high-intensity activities with recovery intervals of unpredictable (rather short) duration (*Dolci et al., 2020*). Thus, team sports players are usually not able to quickly restore the phosphagen system. As a result, they rely more on the glycolytic system. In contrast, endurance training and competition are based more on prolonged efforts below maximum intensity, hence, ATP resynthesis *via* the glycolytic system is less utilized.

## Differences between athletic groups in mechanical variables

In addition, while absolute peak and average power differed significantly between speed-power and endurance athletes, the weight- and SMM-adjusted values showed no differences between the groups. This is consistent with the observations of Davies and Sandstorm (*Davies & Sandstrom, 1989*), who observed that maximal power and capacity were determined by body size and muscularity. This suggests that the relative muscle mechanical efficiency is a rather constant value. Our study shows that the same mechanical efficiency per kg body mass or SMM can be achieved through different contributions of energy systems.

## Training influence

Our data did not clearly indicate that competitive athletes change the proportion of energy systems with ongoing training. Regardless of the categorization applied, athletes maintained

very similar proportions of energy systems between the first and second examinations. Several explanations for this observation are possible. First, one can speculate that after years of specific professional training, and reaching high levels of performance, energy system contributions are finally 'shaped' and do not respond to standard cyclic training stimuli. In support of this view, we observed a statistically significant, however, only a two-percent decrease in aerobic metabolism contribution in our amateur soccer players, but not in our national-level athletes. Research indicates that long-term training leads to specific adaptations that stabilize metabolic and physiological responses due to several factors: genetic limits (*Costa et al., 2012*), long-term specific adaptation to training (*Mujika & Padilla, 2001*), optimization of muscle fiber types depending on the sport (*Ross & Leveritt, 2001*; *Zierath & Hawley, 2004*), and changes in enzymatic activity (*Harmer et al., 2000*; *Hawley, 2002*). For example, endurance athletes maintain their aerobic capacity ($\dot{V}O_2$max) and metabolic efficiency even with changes in training intensity or duration (*Barbieri et al., 2024*). Only minor variations were observed in the physical and competitive performance of top-level sprinters, with no significant changes noted in sprint speed, jumping ability, or peak power output (*Loturco et al., 2023*). It seems that an athlete's energy profile appears to be more resistant to change the longer the sports career and related training stimuli. Perhaps if we examined untrained individuals, the energy system contribution would change significantly with training. This requires further research.

Another explanation of insignificant changes may be the nature of the training phase analyzed, *i.e.*, general preparation period. In most sports, this period is focused on overall physical fitness, strength, and conditioning, with less emphasis on specialized training loads. In particular, endurance and resistance loads are used, rather than speed-power exercises that are the basis for all-out test performance. Thus, the lack of adequate training stimuli would be the reason for the unchanged energy system contribution in our high-level endurance and speed-power athletes. However, our research team demonstrated in previous studies that the blood level of sensitive biomarkers related to energy metabolism (*e.g.*, hypoxanthine, ammonia, and enzyme hypoxanthine-guanine phosphoribosyl-transferase) changed significantly as early as in the general preparation period in highly trained athletes (*Pospieszna et al., 2019*; *Włodarczyk et al., 2020*; *Zieliński & Kusy, 2015*). Thus, it seems unlikely that the multi-week regular training we analyzed in this study did not result in any metabolic adaptations, regardless of the type of load. It seems that both long-term adaptation and the character of the general preparation phase may be responsible for the lack of significant changes in energy systems contribution. Research in the consecutive training phases (especially in the competitive) could better explain the role of the effects of ongoing training.

## Practical applications

Since individual energy metabolism profiles in high-performance athletes may be resistant to change, the picture of energy systems contribution obtained during the specific 15-s all-out test may be used as a supplementary diagnostic tool in assessing one's suitability for a particular sport that best match one's metabolic capabilities. However, the measurement method probably does not allow tracking changes in energy profiles in high-level athletes.

The findings may provide coaches and sport scientists with specific knowledge about classification athletes to choose adequate discipline at an early or later stage of sports training. In addition, through repeated testing on specific athletes, coaches can acquire the ability to estimate an athlete's energy profile. This allows them to identify the optimal performance ratios over the course of an annual training cycle, both in the all-out test and during the sport-specific effort. In addition, one cannot exclude that the energy profile is inherited to some extent and forms the basis of early selection for sports. If so, the energy metabolism profile may be relatively constant throughout the athletic career, and its 'trainability' would be low.

### Limitations and strengths

Given the above, it is also important to note the specificity of the measurement method used. Oxygen uptake and peak lactate levels are the basis for calculating the energy expenditure and contribution of the three metabolic systems. However, these parameters usually do not change considerably with training phases in elite athletes (*Trinschek, Zieliński & Kusy, 2020*; *Włodarczyk et al., 2020*), which is consistent with this study. Possibly, changes in the proportion of energy systems were not observed because these specific parameters did not change significantly in our highly trained groups (Table S1). The inference from our study is limited to male athletic groups in their preparatory period of the annual training cycle. Caution should also be exercised when comparing with other sports disciplines than the ones we analyzed. The interpretation of our findings should certainly not be extrapolated to the general population. In the future, data should be obtained from male and female athletic cohorts during the competitive phase, from athletes of different disciplines and untrained individuals. This could facilitate the identification of an athlete's predisposition and profile, enabling the selection of disciplines that align with their metabolic capabilities. The novelty of this study is the comparison of three homogenous athletic groups with entirely different physiological adaptations, including two highly trained and one amateur group, and the attempt to determine the influence of the training period on energy systems contribution.

## CONCLUSION

In summary, the contribution of energy systems during sprint exercise is related to the sports specialization and the resulting metabolic profile, with the aerobic capacity level being a significant discriminatory factor. The contribution of energy systems does not significantly change in response to the standard training process in highly trained athletes. Our study suggests that it is specific long-term training adaptations that are responsible for the energy systems contribution during sprint exercise in high-level athletic cohorts, while the impact of current training stimuli is heavily limited.

### Abbreviations

| | |
|---|---|
| **ANOVA** | analysis of variance |
| **EPOC** | excess post-exercise oxygen consumption |
| **PCr-LA-O$_2$** | three-component method for calculating the energy systems contribution |

| BM | body mass |
|---|---|
| BF | body fat |
| SMM | skeletal muscle mass |
| LLM | leg lean mass |
| PP | peak power |
| MP | mean power |
| FI | fatigue index |
| $\dot{V}O_2$ | oxygen uptake |
| ATP | adenosine triphosphate |
| AMP | adenosine monophosphate |
| DXA | dual-energy X-ray absorptiometry |
| $E_{PCR}$ | phosphagen system |
| $E_{LA}$ | glycolytic system |
| $E_{AER}$ | aerobic system |

## ACKNOWLEDGEMENTS

We would like to thank the athletes and their coaches for participating in the study.

### Funding

The research was funded by the Poznan University of Physical Education under the Young Scientists Development program, decision issued on April 2021. The funders had no role in study design, data collection and analysis, decision to publish, or preparation of the manuscript.

### Grant Disclosures

The following grant information was disclosed by the authors:
Poznan University of Physical Education under the Young Scientists Development program.

### Competing Interests

The authors declare there are no competing interests.

### Author Contributions

- Damian Archacki conceived and designed the experiments, performed the experiments, analyzed the data, prepared figures and/or tables, authored or reviewed drafts of the article, and approved the final draft.
- Jacek Zieliński performed the experiments, authored or reviewed drafts of the article, and approved the final draft.
- Barbara Pospieszna performed the experiments, authored or reviewed drafts of the article, and approved the final draft.
- Michał Włodarczyk performed the experiments, authored or reviewed drafts of the article, and approved the final draft.
- Krzysztof Kusy conceived and designed the experiments, performed the experiments, analyzed the data, authored or reviewed drafts of the article, and approved the final draft.

## Human Ethics

The following information was supplied relating to ethical approvals (i.e., approving body and any reference numbers):

Ethics Committee at the Poznań University of Medical Sciences

## Data Availability

The raw data is available in the Supplemental File.

## Supplemental Information

Supplemental information for this article can be found online at http://dx.doi.org/10.7717/peerj.17863#supplemental-information.

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
