# Peer review of "The contribution of energy systems during 15-second sprint exercise in athletes of different sports specializations"

_PeerJ, doi:10.7717/peerj.17863_

## Round 0.1 · original submission · Major Revisions

The study entitled “The contribution of energy systems during 15-second sprint exercise in athletes of different sports specializations” demonstrated excellent findings using an appropriate methodological approach. However, some important points must be clarified in the manuscript. Your article has great potential for publication on PeerJ, but the reviewers have requested substantial changes to be made, mainly in methodology and discussion sessions.

Reviewer 1 ·

Basic reporting

PEER J #98854


Congratulations to the authors for their diligent efforts in conducting this study and contributing to the field of Exercise Science. Although the paper is interesting, I have many concerns.

My main concern is the lack a detailed description of the 14-week training program, including exercise types, resistance training specifics, intensity measures, and session durations, is essential for understanding the study's aim to assess long-term physical training effects on energy system contributions

Please ensure that relevant academic sources are used for citations instead of textbooks, such as the one referenced in line 51 (Robergs, 1996). Additionally, please review the list of references for accuracy. For instance, when citing articles, ensure that the correct citation format is used, such as in the case of Bogdanis, G. C. (1998). Instead of 'Article in,' consider using the appropriate format accordingly.

Lines 54-55: please provide reference for "There are three main reactions within the phosphagen system: creatine kinase, adenylate kinase, and adenosine monophosphate deaminase reactions."

Line 57: be more specific than "a few seconds" or provide an example. By the way, the reference cited (Pilegaard et. al.) lacks information on the reference list!

Lines 65-66: I am not sure what you meant by "acute- and prolonged"?

Please ensure consistency in the formatting of measurements throughout the paper. Decide whether to use 100-meters or 100-meter or '100 meters' and '200 meters' or '100 m' and '200 m', and maintain this style consistently across the entire document.

L84: Please ensure consistency in the usage of 'using,' 'about,' or the symbol '~'. I recommend reviewing the paper and employing the symbol for abbreviations where appropriate, as it aligns more consistently with scientific style. Using both the symbol "~" and the word "about" together would indeed be redundant, as they serve the same purpose of indicating an approximate value.

Experimental design

L127-128: use a space to separate km from the numbers (i.e., 1.5 km; 40 km)

L121 For clarity, consider presenting the information about participants in a more organized manner. Using a table to outline the participants in each respective group could enhance readability and comprehension. Additionally, providing details about any exclusions from the initial sample size of 102 participants would improve transparency and understanding of the final sample size used in the study.

L137: please replace "explosive" with power-based, as "explosive" is not a proper term in exercise science.

L145: "The study period covered the general preparatory phase of the individual annual training cycle of each group (14 weeks between November and March)." I recommend avoiding vague terms like 'general preparatory phase,' as they lack clarity. Additionally, please provide clearer justification for the choice of a 14-week duration. Considering that there are ~ 5 months, or ~ 20 weeks, between November and March, it would be helpful to explain why a 14-week duration was selected instead.

L184: please rephrase "load of 25 W" for "power or intensity of 25 W" (see https://www.tandfonline.com/doi/abs/10.1080/02640410601072757). Options: "The protocol began with a 5-min warm-up at each athlete's own pace, during which a power output of 25 W was maintained for one minute". Or "The protocol began with a 5-min warm-up at each athlete's own pace, during which an intensity of 25 W was maintained for one minute.

L262: "In addition to the criterion of sports specialization, participants were categorized into groups based on medians for FGC and PP"..."low-or or high-VO2max and low…" These sentences should be included in the experimental design section for better coherence and understanding.

It's crucial to have a detailed description of the training program followed by participants during the 14-week period. In addition to the number of sessions per week and the duration of each session, it's essential to include precise information about the type of exercises performed, including resistance training details such as weight lifted, number of repetitions, rest intervals, number of sets, as well as intensity measures such as rating of perceived exertion (RPE), heart rate (HR), lactate levels, or percentage of maximum performance. Given that the authors aim to examine the effect of long-term physical training on energy system contributions, a clear presentation of the training programs is necessary.

Validity of the findings

L378-384: Instead of speculating, it's preferable to present evidence-based assumptions or provide references to support the statement. Alternatively, the rationale for the assumption could be strengthened by providing a more robust explanation based on established principles or previous research findings.

The discussion section contains numerous speculative points, which may not adhere to best practices in academic writing.

Reviewer 2 ·

Basic reporting

1. Clear and unambiguous, professional English used throughout:
• The manuscript is written in clear and professional English. The text is unambiguous, and technical terms are used accurately. The overall tone and style conform to professional standards of courtesy and expression. However, a few minor grammatical errors and awkward phrasings can be improved for better readability. For instance, the abstract could be made more concise and direct.
2. Literature references, sufficient field background/context provided:
• The introduction and background sections are comprehensive, providing a solid context for the study. The authors have cited relevant literature extensively, including seminal works and recent studies in the field of exercise physiology and sports science. This helps to situate the current study within the broader body of knowledge and demonstrates its relevance. However, a few more recent studies could be incorporated to reflect the latest advancements in the field.
For Example;
Tortu, E., Hazir, T., Kin Isler, A. (2024). Energy System Contributions in Repeated Sprint Tests: Protocol and Sex Comparison. Journal of Human Kinetics, 92, 87-98. https://doi.org/10.5114/jhk/175862

Professional Article Structure, Figures, Tables, Raw Data Shared
3. Article structure:
• The article follows a standard scientific structure with clearly defined sections: Abstract, Introduction, Methods, Results, Discussion, and Conclusion. Each section is well-organized, making the manuscript easy to follow. Significant departures from this structure are not observed, and the format conforms to the journal's guidelines.
4. Figures and Tables:
• Figures and tables are relevant to the content of the article and are of sufficient resolution. They are appropriately described and labeled, enhancing the reader's understanding of the data presented. However, the legends for some figures could be more detailed to provide better context without referring back to the main text.
5. Raw Data:
• The manuscript mentions that all appropriate raw data have been made available in accordance with the Data Sharing policy. This transparency is commendable and aids in the reproducibility of the research. Supplementary materials, including detailed data sets and additional tables, are provided, ensuring that all relevant information is accessible.
Self-contained with Relevant Results to Hypotheses
6. Self-contained submission:
• The submission is self-contained and presents a coherent body of work. It represents an appropriate unit of publication, including all results relevant to the hypotheses. The study is not inappropriately subdivided to increase publication count, maintaining the integrity of the research.
7. Relevant results:
• All results are directly related to the initial hypotheses. The discussion section thoroughly interprets the findings, placing them in the context of existing literature. The conclusions drawn are logical and highlight the significance of the results in advancing understanding in the field.
Detailed Analysis
• Enhanced Clarity and Readability: The manuscript could benefit from minor revisions to improve clarity and readability. Complex sentences should be simplified, and technical jargon explained where necessary to ensure accessibility to a broader audience.
• Expanded Background and Literature Review: The background section is comprehensive, but the inclusion of a few more recent studies would provide an updated perspective on the field. This would help in highlighting the novelty and importance of the current study.
• Detailed Methodology: The methodology section is detailed and provides a clear understanding of the experimental design, procedures, and data analysis. However, additional information on the rationale behind certain methodological choices would be beneficial for readers unfamiliar with the specific techniques used.
• Comprehensive Data Presentation: Data presentation is clear, with tables and figures complementing the text. Detailed legends and captions for all figures and tables would further enhance understanding.

Experimental design

Experimental Design
1. Original Primary Research Within Aims and Scope of the Journal:

Evaluation: The article presents original primary research that fits well within the aims and scope of the journal focused on sports sciences. The study investigates the contribution of energy systems during a 15-second sprint exercise in athletes from different sports specializations. The research question is clearly defined, relevant, and meaningful. It effectively identifies a knowledge gap in understanding the energy system contributions across various athletic disciplines and how long-term training adaptations influence these contributions.
Revision Suggestion: The introduction should further emphasize the unique contribution of the study and clearly articulate how it addresses the identified knowledge gap. Additional context on the significance of the research question and its implications for the field of sports science would strengthen this section.
2. Rigorous Investigation Performed to a High Technical & Ethical Standard:

Evaluation: The study has been conducted rigorously and meets high technical standards. It follows the ethical guidelines approved by the Poznan University of Medical Sciences Ethics Committee. Participants provided informed consent, and all procedures conformed to the principles of the Helsinki Declaration.
Revision Suggestion: The manuscript should provide more detailed information on the ethical approval process and how ethical standards were maintained throughout the study. This includes a clearer description of participant consent procedures and measures taken to ensure ethical compliance.
3. Methods Described with Sufficient Detail & Information to Replicate:

Evaluation: The methods section is detailed and comprehensive, allowing for reproducibility. It includes descriptions of participant selection, testing protocols, data collection, and analysis methods.
Revision Suggestion: Further explanation is needed for some methodological choices, particularly the rationale behind using the PCr-LA-O2 method for calculating energy system contributions. Additionally, the statistical methods and their application should be elaborated to enhance transparency and reproducibility.
Detailed Revision Requests
Condense and Clarify the Abstract:

The abstract should be more concise and direct. Summarize the main findings and conclusions clearly and succinctly.
Expand the Introduction and Literature Review:

Enhance the introduction with more recent references to underscore the study's relevance and contribution. Provide a more comprehensive literature review to contextualize the research within the existing body of knowledge.
Justify Methodological Choices:

Provide detailed justifications for the selected methodologies, particularly the PCr-LA-O2 method. Explain why these methods are appropriate for the study's objectives and how they compare to alternative approaches.
Enhance Figures and Tables:

Improve the clarity of figure and table descriptions. Each figure and table should be fully understandable without referring back to the main text.
Expand the Discussion of Results:

The discussion section should provide a deeper analysis of the results. Compare findings more extensively with existing literature and elaborate on the study's scientific contributions.

Validity of the findings

1. Impact and Novelty Not Assessed; Meaningful Replication Encouraged Where Rationale & Benefit to Literature Is Clearly Stated:

Evaluation: The study does not focus on the novelty or the broader impact of the findings, which aligns with the journal's guidelines. The research question is well-defined and addresses a meaningful gap in the literature regarding how long-term training adaptations affect energy system contributions during sprint exercises in athletes of different specializations. The rationale for this study is clearly stated, emphasizing the importance of understanding specific metabolic adaptations in high-performance athletes.

Revision Suggestion: While the manuscript effectively outlines the rationale for the study, it would benefit from a clearer discussion on the potential implications of these findings for future research and practical applications in sports training and performance. This could include suggestions for how these results could inform training strategies and athlete management.

2. All Underlying Data Have Been Provided; They Are Robust, Statistically Sound, & Controlled:

Evaluation: The manuscript provides detailed information on the methods and the data collected. The data appear to be robust, statistically sound, and well-controlled. The use of the PCr-LA-O2 method for calculating energy system contributions, along with comprehensive statistical analyses, supports the validity of the findings.

Revision Suggestion: Ensure that all raw data are made available in a publicly accessible repository, as this will enhance the transparency and reproducibility of the research. Additionally, providing more details on the data cleaning and preparation processes would strengthen the methodological rigor.

3. Conclusions Are Well Stated, Linked to Original Research Question & Limited to Supporting Results:

Evaluation: The conclusions are appropriately stated and linked to the original research question. They are limited to the results obtained, avoiding overgeneralization. The findings are interpreted in the context of the study's hypotheses, supporting the notion that long-term training adaptations rather than ongoing training significantly influence energy system contributions in high-performance athletes.

Revision Suggestion: The discussion section should be expanded to include a more in-depth analysis of the implications of the findings. Specifically, the authors should address the potential mechanisms underlying the observed stability in energy system contributions despite ongoing training. Additionally, the limitations of the study should be discussed more thoroughly, including potential sources of bias and the generalizability of the findings to other populations or sports.

Detailed Revision Requests

Elaborate on the Study's Implications:

Provide a more comprehensive discussion on how these findings could influence future research directions, training practices, and athlete management strategies. Highlight specific applications of the results in sports science and coaching.

Expand the Discussion on Methodological Rigour:

Include additional information on data cleaning, preparation, and any assumptions made during analysis. This will help readers assess the robustness of the findings and the reliability of the methods used.
Address Study Limitations in Greater Detail:

Discuss the limitations of the study in more detail. This should include potential biases, the representativeness of the sample, and the extent to which the findings can be generalized to other populations or settings.

Clarify Potential Mechanisms:

Explore potential mechanisms that could explain the stability of energy system contributions despite ongoing training. This could include physiological or biochemical factors that contribute to the observed patterns.

Additional comments

Providing a more detailed justification for the chosen duration of the Wingate test.

Clarity and Readability:
The manuscript is generally well-written, but a few sections could benefit from clearer and more concise language. Simplifying complex sentences and reducing technical jargon where possible will enhance readability and accessibility for a broader audience.
Ensure that the abstract provides a succinct summary of the main findings and their significance without excessive detail. This will help readers quickly grasp the core message of your study.

Structure and Flow:
The structure of the manuscript follows a logical progression, but transitions between sections could be smoother. Consider adding brief introductory sentences or paragraphs to seamlessly connect different parts of the manuscript.
The results section is comprehensive but could be more reader-friendly with the inclusion of summary tables or bullet points highlighting key findings.

Figures and Tables:
Figures and tables are crucial for illustrating your results. Ensure that all figures and tables are of high quality and clearly labeled. Legends should be detailed enough to make the figures understandable without referring back to the text.
Consider adding more visual aids, such as flowcharts or diagrams, to explain complex processes or methodologies.

Discussion and Implications:
While the discussion is thorough, it would benefit from a more explicit connection to practical applications. Discuss how your findings can be applied in real-world sports training and performance contexts.
Highlight any potential implications for different types of athletes and how coaches and trainers can use this information to optimize training programs.

Limitations and Future Directions:
You have mentioned some limitations of your study, but a more detailed discussion on this topic would strengthen your manuscript. Address any potential biases, the representativeness of your sample, and the generalizability of your findings.

Suggest specific areas for future research based on your results. This will provide a clear path forward for other researchers interested in this field.

Reviewer 3 ·

Basic reporting

The manuscript is very well written and adheres to the journals requirements.

Experimental design

Participants
Line 124 please include 1 sentence to describe in more detail what is meant by ‘training specificity’ ? Advise on training load and type completion . This is mentioned on Lone 152-157 but please advise on how much anaerobic training each athlete group completed

Line 133 ‘mixed group’ change to ‘team sport athletes’? This group is also ‘amateur’ - what level of completion were the other athletes ‘national ‘ , ‘international ‘ or world / Olympic standard ?

Study design
Line 145 -146 has been repeated. Delete

Body composition
Please describe in specific detail as to the subject preparation for the DXA scan. This is important as pre scan treatment is an important factor influencing results of the scan.
Who completed the scan? What was their training and certification?

Oxygen uptake and lactate
Line 199 change the sub heading to ‘exercise testing ‘
Line 209 change ‘ergospirometric ‘ to ‘incremental exercise test to exhaustion’

Results
Body composition and aerobic capacity
Line 277-286 plus the entire results section. In this section please include mean and standard for each variable of interest for each group

Validity of the findings

Discussion

Line 320-322
It is stated that long term adaptations rather than training stimuli determine energy system contribution. But in fact, you have not recorded and monitored any training . Please review and modify the wording of this statement.

Line 326
The standard ‘Wingate’ test is 30 sec duration. Please use the wording ‘all out’ test or ‘anaerobic ‘ test. Then state the duration of the test .


Line 340 - 365 discussion of no differences in energy contribution to exercise
Was this due to the training phase each athlete was within? Having information about the intensity of training would be useful. The fact the all out test was conducted in cycling and not running may also be a factor as all athletes predominantly run , with exception of triathletes who cycle. The mode of exercise that each athlete trains in may be a factor.

Additional comments

This is an interesting study that uses all out exercise testing with metabolic analysis together with estimation of energy contribution to exercise to determine differences in a anaerobic performance and response in different groups of athletes (endurance , sprint and team sport) . The manuscript is well written and of interest to sport scientists
The limitation of this study are that the analysis (and conclusions made) was conducted without any report of training characteristics of the athletes, especially anaerobic training. The testing was done in cycling , which is unfamiliar for most of the athletes i.e. they don’t train in cycling except the endurance athletes (triathletes ).
The performance status of the athletes is unknown - at what level where they competing ?

---

## Round 0.2 · accepted · Accept

Dear Author,

Congratulations! After your diligent work addressing the reviewers' comments, I am pleased to inform you that your manuscript has been accepted for publication in PeerJ. This version is more concise and formal, enhancing clarity and flow.

Reviewer 2 ·

Basic reporting

The manuscript titled "The contribution of energy systems during 15-second sprint exercise in athletes of different sports specializations" provides a clear and comprehensive analysis of the energy system contributions during sprint exercises in athletes with varying sports specializations. The authors have successfully utilized professional English throughout the manuscript, ensuring clarity and precision in their reporting.

Introduction and Background
The introduction and background sections effectively set the context for the study. The authors provide a thorough review of the relevant literature, highlighting the significance of understanding energy system contributions in different types of athletes. The rationale behind the study is clearly articulated, and the research questions are well-defined and relevant.

Literature Referencing
The manuscript is well-referenced, drawing on a wide range of studies to support the discussion. The references are relevant and current, providing a solid foundation for the research. This comprehensive referencing not only supports the authors' arguments but also situates the study within the broader context of existing research.

Structure and Presentation
The structure of the manuscript conforms to PeerJ standards, and the discipline norms are adhered to. The manuscript is well-organized, with each section flowing logically into the next. The use of subheadings enhances readability and allows for easy navigation through the text. This structure facilitates a clear presentation of the methods, results, and conclusions.

Figures and Tables
The figures and tables included in the manuscript are of high quality and are appropriately labeled and described. They effectively illustrate the key findings and complement the text. The visual aids are relevant and enhance the reader's understanding of the data presented.


In conclusion, the manuscript meets the basic reporting standards required for publication.

Experimental design

Original Research Scope
The study addresses a well-defined and relevant research question, focusing on the contribution of different energy systems during sprint exercises in athletes from various sports specializations. This research fills a significant knowledge gap in sports science, particularly in understanding the metabolic demands of sprint exercises across different athletic disciplines.

Methodological Rigor
The investigation is carried out with rigorous methodological precision. The authors describe the procedures in sufficient detail, ensuring that the study can be replicated by other researchers. The inclusion of a well-defined sample of endurance, speed-power, and mixed-sport athletes allows for meaningful comparisons and enhances the validity of the findings.

Data Collection and Analysis
The methods of data collection, including the use of the PCr-LA-O2 method for calculating energy contributions and the comprehensive approach to body composition analysis, are described thoroughly. The statistical analyses are appropriate and robust, providing a reliable basis for the study's conclusions.

In summary, the experimental design of this study is exemplary, demonstrating rigorous scientific methodology and ethical integrity. The detailed and systematic approach ensures that the findings are both valid and replicable, making a valuable contribution to the field.

Validity of the findings

The findings of the study are well-supported by robust data and sound statistical analysis. The results clearly link to the original research questions and are presented in a logical and coherent manner. The study's conclusions are firmly grounded in the data, reflecting accurate and reliable insights into the energy system contributions during sprint exercise. Overall, the validity of the findings is strong, providing meaningful contributions to the field of sports science.

Additional comments

No comments